# Type 1 diabetes mellitus and educational attainment in childhood: a systematic review

Natalie Jayne Oakley ,[1] Dylan Kneale ,[2] Mala Mann,[3] Mariann Hilliar,[4] Colin Dayan,[5] John W Gregory,[1] Robert French [5,6]

[1]Division of Population Medicine, School of Medicine, Cardiff University, Cardiff, UK
[2]Social Science Research Unit (SSRU), University College London, London, UK
[3]Specialist Unit for Review Evidence, Cardiff University, Cardiff, UK
[4]School of Medicine Library, Cardiff University, Cardiff, UK
[5]Division of Infection and Immunity, School of Medicine, Cardiff University, Cardiff, UK
[6]Centre for Multilevel Modelling, Graduate School of Education, University of Bristol, Bristol, UK

**Correspondence to**
Dr Robert French;
frenchr3@cardiff.ac.uk

## ABSTRACT

**Objectives** The primary objective of this systematic review was to evaluate available literature on whether type 1 diabetes mellitus (T1DM) has an impact on educational attainment in individuals undertaking high stakes standardised testing at the end of compulsory schooling.

**Design** A systematic review was undertaken comparing educational attainment for individuals with and without T1DM who have undertaken high stakes testing at the end of compulsory schooling.

**Data sources** A comprehensive search of MEDLINE, MEDLINE (epub ahead of print, in-process and other non-indexed citations), EMBASE, Web of Science, British Education Index, Education Resources Information Center and Cumulative Index to Nursing and Allied Health Literature was undertaken on 15 January 2018 and updated on 17 January 2019.

**Eligibility criteria** Included studies fulfilled the following criteria: observational study or randomised controlled trial; included individuals who have undertaken high stakes testing at the end of compulsory schooling; compared the grades obtained by individuals with T1DM with a representative population control.

**Data extraction and synthesis** Two reviewers performed study selection and data extraction independently. Quality and risk of bias in the observational studies included were assessed using the Newcastle-Ottawa Scale. A detailed narrative synthesis of the included studies was completed.

**Results** 3103 articles were identified from the database search, with two Swedish cohort studies (using the same linked administrative data) meeting final inclusion criteria. A small but statistically significant difference was reported in mean final grades, with children with T1DM found to have lower mean grades than their non-diabetic counterparts (adjusted mean difference 0.07–0.08).

**Conclusions** More contemporary research is required to evaluate the impact of T1DM in childhood on educational attainment in individuals undertaking high stakes standardised testing at the end of compulsory schooling, taking into consideration the substantial advances in management of T1DM in the last decade.

**PROSPERO registration number** CRD42017084078.

## INTRODUCTION

Type 1 diabetes mellitus (T1DM) is an autoimmune disorder which destroys the insulin-producing beta cells in the pancreas,

### Strengths and limitations of this study

► Previous studies have found an association between type 1 diabetes mellitus (T1DM) in childhood and reduced cognitive function, however, there is less evidence assessing the effect of T1DM on educational attainment, particularly in the form of high stakes examinations.

► This review has comprehensively evaluated available literature reporting the impact of T1DM on educational attainment in individuals undertaking high stakes standardised testing at the end of compulsory schooling.

► The findings were reported using the Cochrane Handbook for Systematic Reviews of Interventions and recommended methods and checklist of the Preferred Reporting Items for Systematic Reviews and Meta-Analyses, with the quality of evidence assessed using the Grading of Recommendations Assessment, Development and Evaluation system.

► Study selection and data extraction were performed independently by two reviewers, and the quality and risk of bias of the studies included in this review were assessed using the Newcastle-Ottawa Scale.

► Limitations of this review include the use of narrow inclusion thresholds, commensurate with the research objectives, resulting in a small number of included studies thereby precluding the use of meta-analysis and limiting generalisability of the results.

preventing the body from adequately regulating blood glucose levels.[1] According to the 2017 International Diabetes Federation Diabetes Atlas eighth edition, the number of children diagnosed with type 1 diabetes is increasing annually, with an estimated 96 100 new cases worldwide every year.[2]

Physiological complications of diabetes can include hypoglycaemia, hyperglycaemia and ketoacidosis.[1] These complications, especially if recurrent, have the potential to impact on educational attainment via a combination of mechanisms including altered cognitive function and non-attendance for acute treatment.[3 4] Glucose is the principal energy source of the human brain. Hypoglycaemia,

particularly if severe or recurrent, can result in neuroglycopenia which, in turn, may result in neuronal injury and subsequent cognitive impairment.[5] Hyperglycaemia, particularly when associated with ketoacidosis, has also been found to have detrimental effects on the brain via damage to white matter, disrupted functioning of the blood-brain barrier and transient focal cerebral oedema. These changes can again impair cerebral functioning resulting in cognitive impairment.[6] There is also the potential for attendance and attainment to be affected by hospital appointments or illness at critical points in the educational trajectory.[7] This is especially important when considering high stakes testing at the end of compulsory schooling which has the potential to significantly impact future social and economic outcomes, such as social status, employment and income.[8]

Broadly, high stakes testing can be defined as examinations or assessments which carry serious and important consequences for the individuals taking the tests. In the case of high stakes tests at the end of compulsory schooling, these important consequences may include subsequent educational and employment prospects and opportunities. High stakes tests are often 'standardised', meaning there are specific rules and regulations involved in providing and completing the test. This aims to ensure every individual taking the test receives the same resources and directions, allowing a more accurate interpretation of individuals' performance and ability.[9]

Many patients and their families report worries about the support available for children with diabetes at school and concerns about the potential negative impact that T1DM may have on school attendance.[4 7] In an observational study published in 2007, Amillategui et al utilised self-reporting questionnaires to identify specific parental concerns regarding T1DM in the school setting. Many of these concerns focused on aspects such as glycaemic control, social integration, medication administration and understanding of staff members, which may affect children's attainment, well-being and experience in school.[10]

Previous literature has studied the effects of T1DM on cognitive functioning in children. In a review published in 2004, Desrocher and Rovet found an association between T1DM in childhood and poorer neurocognitive outcomes, with the most significant deficits found to be related to younger age of disease onset, episodes of hypoglycaemia and hyperglycaemia around the time of puberty.[11] In 2008, Gaudieri et al reported that T1DM in childhood was associated with poorer performance in learning and reduced memory skills.[12] Further studies published by Naguib et al in 2009[13] and He et al in 2018[14] reported similar findings, suggesting a detrimental impact of T1DM in childhood on cognitive function. However, there is less evidence as to the effect T1DM has on educational attainment, particularly in the form of high stakes examinations, and the magnitude of this impact.[7]

Each of the four home nations within the UK (England, Northern Ireland, Scotland and Wales) has specific legislation focused on support for children and young people with additional learning needs and medical conditions including T1DM in school.[15–20] Quantifying the impact of T1DM on educational attainment at the end of compulsory schooling and identifying domains most affected may be useful in assessing what and how much support should be focused on children with T1DM in school.

The primary objective of this review was to identify and analyse the available literature on whether T1DM has an impact on educational attainment in individuals undertaking high stakes standardised testing at the end of compulsory schooling.

Secondary objectives included assessment of the effect of T1DM on school attendance and educational attainment at other stages on the educational trajectory if reported.

## METHODS

The protocol for this systematic review is registered with PROSPERO (International Prospective Register of Systematic Reviews) at the National Health Service Centre for Reviews and Dissemination at the University of York[21] and has been previously published.[22] The Cochrane Handbook for Systematic Reviews of Interventions[23] and recommended methods and checklist of the Preferred Reporting Items for Systematic Reviews and Meta-Analyses[24] were used to structure this systematic review.

Studies were included if they fulfilled the following inclusion criteria: (1) observational study (including prospective and retrospective cohort and case-control studies) or randomised controlled trial; (2) included individuals who have undertaken high stakes testing at the end of compulsory schooling; (3) compared the grades obtained by individuals with type 1 diabetes with a representative population control. The primary outcome assessed was grade obtained following high stakes testing at the end of compulsory schooling. Secondary outcomes were school attendance and grades obtained at other stages on the educational trajectory.

Comprehensive electronic literature search strategies were developed to search the following seven databases: Ovid MEDLINE (1946–present), Ovid MEDLINE (epub ahead of print, in-process and other non-indexed citations), Ovid EMBASE (1947–present), Thomson Reuters Web of Science, EBSCO British Education Index, EBSCO Education Resources Information Center and EBSCO Cumulative Index to Nursing and Allied Health Literature. Only studies published since January 2004 in the English language were considered. The initial search was undertaken on 15 January 2018 and then updated on 17 January 2019 to identify any further studies. The results for the initial and updated searches were combined. The search strategies used for the above databases are given in online supplementary appendix 1.

To identify (1) studies in progress, (2) unpublished research, or (3) research reported in the grey literature, the electronic Table of Contents of key journals (listed in online supplementary appendix 2) and trial registries[25–27] were searched for studies published within the last 2 years. Finally, additional searches for studies were undertaken via relevant websites including Diabetes.org.uk and jdrf.org.uk, review articles, reference lists and citation tracking of included studies.

After removal of duplicates, two independent reviewers (NJO and RF) screened study titles and abstracts using Eppi-Reviewer V.4.0.[28] Full texts of potentially eligible records were then screened according to set inclusion criteria. The rationale for the exclusion of studies at each stage was documented. The remaining included studies underwent data extraction by NJO and RF independently.

The quality and risk of bias in the observational studies included in this review were assessed by NJO and RF independently using the Newcastle-Ottawa Scale (NOS).

The principle outcome measure used in included studies was the 'mean difference' between school grades attained. When assessing the likelihood of achieving different grades, the principle binary outcome measure used was the 'OR'. Meta-analysis was not possible because of the substantial duplication of data linkage from the databases, with both papers including a cohort of participants born 1973–1978. As a result, a detailed narrative synthesis of the included papers was undertaken.

The quality of evidence for the primary outcome was assessed using the Grading of Recommendations Assessment, Development and Evaluation (GRADE) system.[29] Results are presented in table 1 as recommended by the Cochrane Handbook for Systematic Reviews of Interventions.[23]

### Patient and public involvement

Patients and the public were not involved in the development and completion of this systematic review.

### RESULTS

The database search identified 3103 papers. After duplicates were removed, 2304 articles underwent abstract screening (figure 1). A total of 45 papers were shortlisted for full-text review. No full text was available for nine studies, and further information was required for one study. The authors of these studies were contacted, but as no response or clarification was obtained, these studies were excluded.

Table 2 lists the reasons for exclusion for the studies rejected on full-text review (full citations listed in online supplementary appendix 3). The abstracts of four additional studies identified from the reference lists of the studies shortlisted for full-text review were screened and rejected.

Two papers met all of the inclusion criteria, both electronic population cohort studies from Sweden,[3 30] as detailed in table 3. The most recent sought to improve on the analysis of the earlier paper by (1) using different methods, (2) using more refined model specifications and (3) adding later life outcomes. Both papers compared mean attainment on a five-point scale[1–5] at age 16 years for those with diabetes to those without diabetes. The sample size decreased for the later study because they used more restricted birth cohorts to allow comparisons of labour force outcomes aged 29 years.

The primary outcome in these two studies was mean final grade across all school subjects, from compulsory secondary school (table 4). Persson stated that these grades were teacher-rated, therefore potentially introducing bias if teachers subconsciously adjusted grades because of diabetes status. For the later exam years (1998–2003) used by Dahlquist, the outcome scale was changed and estimates for this period were given separately by subject only. Both papers showed a similar negative effect of diabetes on mean attainment, after adjusting for confounders. Dahlquist estimated children with T1DM achieved 0.08 fewer final mean marks, while Persson estimated children with T1DM achieved 0.07 fewer final mean marks. Persson also used quantile regression to show that the effect of diabetes was strongest in the lowest quantile of attainment.

Both studies also analysed the mean attainment score in four core subjects—Maths, Swedish, English and Sports. In Sports/Athletics, children with T1DM were found to be more likely to achieve lower grades and less likely to achieve higher grades. The quality of evidence for these outcomes, assessed using the GRADE system,[29] is presented in table 1.

Both studies estimated the effect of age of diagnosis using further models, suggesting lower grade attainment with earlier diagnosis. Dahlquist estimated conditional means for age groups of diagnosis (<2, 2–5, 5–10, 10–15 years), reporting the lowest mean grades in children diagnosed before 2 years of age (2.97), with means of 3.08–3.17 in categories of older age at diagnoses. However, the differences between age groups was found to be not significant. Persson used those without diabetes as the reference group, showing a negative association between earlier diagnosis and attainment. Those diagnosed aged 10–15 years were found to have 0.06 lower attainment, those diagnosed aged 5–9 years were found to have 0.07 lower attainment and those diagnosed aged 0–4 years were found to have 0.15 lower attainment.

The quality and risk of bias for the included studies was assessed as high-quality methodology with low risk of bias using the NOS. Full NOS ratings can be seen in online supplementary appendix 4. Persson scored full marks across the three domains. Dahlquist lost one mark on the outcome assessment domain for not reporting the numbers lost due to linkage (although they did consider the reasons why cases may not be included). Despite both papers being judged to be methodologically sound overall, the certainty of the evidence was deemed to be very low, mainly reflecting the small number of studies identified and available for analysis.

**Table 1** Summary of findings table

The association between T1DM and educational attainment in childhood

Patient or population: individuals who have undertaken high stakes testing at the end of compulsory schooling, under 18 years of age.
Setting: school.
Intervention: known diagnosis of T1DM.
Comparison: no known diagnosis of T1DM.

| Outcomes | Impact | Participants (studies) (n) | Certainty of the evidence (GRADE) |
|---|---|---|---|
| Mean final grade of all school subjects from compulsory schooling at age 16 (Mean final grade) | Both papers found significantly lower attainment of between 0.07 and 0.08 marks achieved by children with T1DM compared with their non-diabetic counterparts. | (Two observational studies) | ⊕◯◯◯ VERY LOW*†‡§ |
| Mean final grade of specific subjects (Maths, Swedish, English and Sports) from compulsory school at age 16 | **Dahlquist and Källén** tested for differences in the odds for each level of attainment by subject in the raw categories and in an alternative four categories where the raw numeric scores (1988–1997) were manipulated to have the same number of categories as the alphabetical marks (1998–2003). In Sports, children with T1DM had a greater odds of a low score and a lower odds of a high score. In Maths and Swedish, children with T1DM had a higher odds of a low score (no clear differences for high scores), and there were no clear effects for English.<br>**Persson _et al_** showed a similar pattern, though all the comparisons of levels within subjects were statistically significant (though comparison for the mean attainment scores for Maths was not significant), with the largest differences in attainment for Sports and the smallest differences for Maths. | (Two observational studies) | ⊕◯◯◯ VERY LOW*†‡§ |

GRADE Working Group grades of evidence
**High certainty:**
We are very confident that the true effect lies close to that of the estimate of the effect.
**Moderate certainty:**
We are moderately confident in the effect estimate: the true effect is likely to be close to the estimate of the effect, but there is a possibility that it is substantially different.
**Low certainty:**
Our confidence in the effect estimate is limited: The true effect may be substantially different from the estimate of the effect.
**Very low certainty:**
We have very little confidence in the effect estimate: The true effect is likely to be substantially different from the estimate of effect.

*Both included studies control for gender, academic year and maternal education. Dahlquist and Källén additionally condition on maternal age and parity, while Persson _et al_ additionally control for paternal education, long-term parental income and parental country of origin.
†Heterogeneity could not be reliably assessed as only two papers met final inclusion criteria (one study with updated methodology). As a result, there is a substantial amount of duplication of cases from the databases, with both papers including a cohort of participants born 1973–1978.
‡Publication bias could not be reliably assessed as only two papers met final inclusion criteria (one study with updated methodology).
§Total number of participants unable to be combined because of the unknown quantity of data duplication from the same cohort of participants born 1973–1978. (With T1DM—Dahlquist n=5159 and Persson n=2392. Without T1DM—Dahlquist n=1 330 968, Persson n=9563.)
GRADE, Grading of Recommendations Assessment, Development and Evaluation; T1DM, type 1 diabetes mellitus.

## DISCUSSION
### Summary of evidence
This systematic review identified two papers (using overlapping Swedish administrative cohorts) assessing the association between T1DM in childhood and educational attainment following high stakes testing at the end of compulsory schooling. Both papers found statistically significant lower mean attainment of between 0.07 and 0.08 marks in those with T1DM. Persson checked this relationship by attainment quantile and showed that the differences were

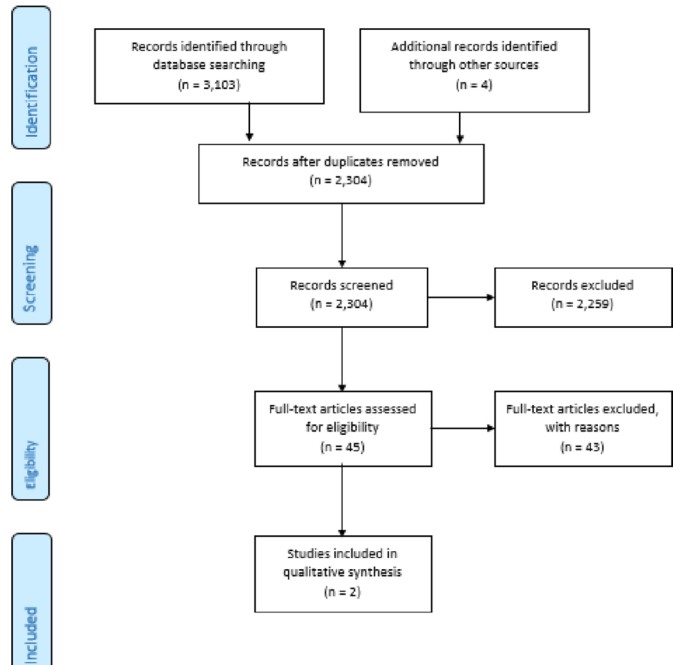

**Figure 1** Preferred Reporting Items for Systematic Reviews and Meta-Analyses 2009 flow diagram (adapted from Moher et al[24]).

greater for lower attainment levels. Both papers considered within-subject attainment and found a negative effect of T1DM. This effect was particularly pronounced in Sports/Athletics but less clear in Maths. It is of great concern that the largest negative association was seen in Sports/Athletics because exercise is a key tool for managing blood sugar and suggests other potential confounding factors such as parental or teacher anxiety regarding the involvement of diabetic children in physical education at school and the risk of hypoglycaemia. Both papers suggested that the negative effect of diabetes on mean attainment was greater for earlier diagnosis, though these effects were only found to be significant in Persson. This hints that the mechanisms driving the differences are not equal for all children with diabetes. Persson also found that the differences in attainment between those with and without diabetes are greater for low ability students.

## Limitations
### Limitations of included studies
With only two papers based on overlapping datasets, it is difficult to generalise to other populations, particularly when the findings of the included studies relate to a single country with a particularly high prevalence rate of T1DM and excellent diabetes-related outcomes.[2]

The included studies controlled for demographics and socioeconomic characteristics, but there are other unobserved covariates which should be considered. Other studies considered living arrangements, country of birth of the child, main language spoken at home, ability to speak English, attendance in preschool and parents' occupation.[31] While the included studies were able to adjust for age of diagnosis, they did not have data for other clinical aspects of diabetes, particularly the quality of diabetes management as measured by HbA1c. Cooper et al[7] were able to show that

**Table 2** Primary reasons for exclusion of studies following full-text review

**Reason for exclusion**

| No final school-based grades assessed | No full text available | Study not specifically assessing T1DM | Not assessing grades following high stakes tests at end of compulsory schooling | No comparison between children with and without T1DM | Study design not in scope |
|---|---|---|---|---|---|
| Bass L 2011<br>Bjerkedal T et al 2006<br>Champaloux S et al 2015<br>Chaudhry T et al 2013<br>Compas B et al 2017<br>Elrayah H et al 2005<br>Gonzalez A et al 2012<br>Hassan M et al 2017<br>Liaqat A et al 2017<br>Maslow G et al 2011<br>Maslow G et al 2012<br>Merrick H et al 2015<br>Neves C et al 2013<br>Nielson H et al 2016<br>Ovesen L et al 2015<br>Ryan C 2012<br>Schiel R et al 2016<br>von Stumm S et al 2011<br>Wennick A et al 2011 | Abusrewil S 2013<br>Catalano D et al 2004<br>Jesic M et al 2013<br>Lynch P et al 2004<br>Milovanovic I et al 2012<br>Mitosi N et al 2013<br>Roman R et al 2016<br>Steen Carlsson K et al 2015<br>Tahirovic H et al 2013 | Almqvist C et al 2016<br>Bezerra A et al 2012 | Cooper M et al 2016<br>Crump C et al 2013<br>Erkolahti R et al 2005<br>Lansing A et al 2018<br>Meo S et al 2013<br>Nasuuna E et al 2016<br>Roman R et al 2017 | Engelke M et al 2008<br>Winnick J et al 2011<br>Bortes C et al 2018 | Fraser A et al 2012<br>Jameson P et al 2006<br>Taras H et al 2005 |

T1DM, type 1 diabetes mellitus.

**Table 3** Characteristics of included observational studies

| Author | Design | Country | Cases (known T1DM) (n) | Controls (no T1DM) (n) | Matching completed (Y/N) | Controlled confounders |
|---|---|---|---|---|---|---|
| Persson *et al*[3] | Electronic population cohort | Sweden | 2392 | 9563 | Y | Year of birth, gender, school year, level of parental education, parental income, parental country of origin |
| Dahlquist and Källén[30] | Electronic population cohort | Sweden | 5159 | 1 330 968 | N | Year of birth, gender, maternal age, parity and educational level |

T1DM, type 1 diabetes mellitus.

a higher HbA1c (mean HbA1c during the 2 years before testing) was associated with poorer attainment, while history of severe hypoglycaemia or history of diabetic ketoacidosis showed no effect on educational attainment.

Finally, neither of the included studies included the secondary outcome, school attendance. In an excluded study, Cooper *et al*[7] showed significantly lower school attendance in children with type 1 diabetes in all age groups (p<0.001); from 2.6% lower attendance in year 3 up to 3.4% lower attendance in year 5. This study and that of Crump *et al*[32] also used attendance as a covariate in models of attainment.

### Limitations of review

The primary limitation of the review was that the thresholds for inclusion were very narrow, meaning that only two studies met the inclusion criteria. The small number of studies (with overlapping individuals) meeting the

inclusion criteria precluded the use of meta-analysis or other evidence synthesis techniques. While the use of such narrow inclusion criteria was commensurate with the research aim, as well as being consistent with the protocol for the review,[22] future systematic reviewers may consider the way in which the following criteria limited the number of studies included.

The first limiting restriction was including only studies modelling attainment from 'high stakes testing at the end of compulsory schooling'. Although this is justified through the impact of these tests on adult life chances, this criteria excluded studies which have otherwise suitable study designs for exploring the association between diabetes and educational outcomes, such as Cooper *et al*.[7] This restriction also excluded studies which lack national standardised testing or administrative data records, either because they considered cohorts prior to such testing

**Table 4** Effect of T1DM on mean final grade of (1) all school subjects and (2) specific subjects (Maths, Swedish, English and Sports) at the end of compulsory school

| | Dahlquist and Källén | Persson *et al* |
|---|---|---|
| (1) Mean final grade of all school subjects from compulsory schooling at age 16 (mean final grade) Additional statistical analysis performed | T1DM: 3.15 (adjusted) No T1DM: 3.23 (adjusted) Mean difference (adjusted): 0.08. T1DM found to have negative impact on mean grade attainment of all school subjects. Mean t value −0.24±0.04. t* value −5.19 p<0.001. | T1DM: 3.13±0.75 (unadjusted) No T1DM: 3.21±0.72 (unadjusted) Mean difference (adjusted): 0.07±0.02, p<0.001. T1DM found to have negative impact on mean grade attainment of all school subjects. Quantile regression—negative effect of T1DM greatest in the lowest quantile of attainment, diminishing with increasing attainment quantiles, becoming statistically insignificant for the highest attaining quantile. |
| (2) Mean final grade of specific subjects (Maths, Swedish, English and Sports) from compulsory school at age 16 | ORs used to compare attainment of each grade in specific subjects in order to estimate effect of T1DM. ▶ Sports/Athletics: clear negative association seen between T1DM and grade attainment. ▶ English: similar pattern to Sports/Athletics however weaker association. ▶ Maths and Swedish: children with T1DM found to be more likely to achieve lower grades however no clear differences were seen for high grades. | Mean difference in attained grades (unadjusted), predicted probability of achieving grades 1–5 in specific subjects, and odds ratios using ordered logistic regression used to estimate effect of T1DM. ▶ Sports/Athletics: clear negative association seen between T1DM and grade attainment. ▶ Similar trend seen in all subjects—largest differences in attainment seen in Sports/Athletics, smallest differences seen in Maths. |

T1DM, type 1 diabetes mellitus.

regimes[33] or were studies from countries without such testing regimes.[34 35]

The second limiting restriction was excluding publications before 2004 in view of changes in the treatment of T1DM. The 2015 National Institute for Health and Care Excellence guidelines[36] state that since 2004 there have been major changes in routine management of type 1 diabetes. While this is pertinent to the UK and most western countries, this date was perhaps less appropriate for low-income and middle-income countries. In addition, studies published since 2004 were based on cases from earlier years before those changes, for example, the cohorts for the included studies in this review were born 1972–1985. This may impact the current applicability of conclusions drawn from these studies, as children with T1DM are likely to now have very different treatment regimens. Both included studies also highlight the change in the school marking system in Sweden in 1998 from a five-level numerical scale to a four-level alphabetical system. Changes to other general examination conditions over this time should also be considered, both in terms of national standardised testing and the use of administrative data records. This shows that more contemporaneous studies are required to assess the impact of advances in treatment alongside current examination conditions and data records. For example, Cooper et al[7] includes data for children born up to 2003, and so can report insulin pump therapy being associated with better school performance than two times per day injections.

The third limiting restriction was requiring representative control cases which allow estimates that are generalisable to wider populations, for example, excluding studies recruiting from a single clinic. As for the restriction on the outcome measure, this excludes countries without registries or linked data. This may lead to biases when the countries with high quality linked data, such as the Nordic countries, are also those with high diabetes prevalence.

One related consideration is the use of matching or population controls. In the two included studies these methods are used without justification of the choice or the matching specifications such as the number of controls per case. Matching is traditionally done to make the treatment and control groups more similar. However, these papers do not provide any evidence or reason why those children with type 1 diabetes would be substantively different in terms of other factors which may influence educational attainment. One motivation for the matching approach is to give the appearance of having well-matched healthy controls (regardless of the sparsity of variables used in the process). However, other diagnosed (or undiagnosed) comorbidities within this cohort may affect educational attainment and should be taken into consideration. For example, Meo et al[35] exclude cases with a range of health conditions (reported tobacco use, gross anaemia, tuberculosis, rheumatic fever, rheumatoid arthritis, vision problems, hearing problems or behavioural problems, use of any medication or hospital admissions other than for diabetes mellitus) to reduce the effect of other comorbidities within the cohort. An alternative to matching with healthy controls is to compare attainment for children with T1DM with children with other health conditions, in order to evaluate the mechanisms through which diabetes affects attainment. For example, asthma may affect sleep and concentration, but have less of an impact on attendance, while epilepsy may share some elements of the social stigma, but have a lower effect on functioning at school.

## CONCLUSION

The studies included in this systematic review show a weak negative association between diabetes and overall grade attainment. In specific subjects, the difference in attainment between children with and without T1DM was found to be greatest in Sports/Athletics and smallest in Maths. The negative effect of T1DM was found to be greatest in the lowest quantile of attainment, and findings also suggested lower grade attainment was associated with earlier age at diagnosis. The trend of increasing prevalence of T1DM makes the need for more robust evidence more pressing. More studies from a broader range of contexts are required, including studies utilising alternative methods to provide evidence from countries where large-scale data linkage is not a viable approach. In addition, studies using more contemporaneous data, acknowledging the substantial advances in management of T1DM in the last decade, and ideally linking clinical data such as HbA1c with educational outcomes, are required to aid future recommendations relating to support at school for children with T1DM, especially in the setting of high stakes testing at the end of compulsory schooling.

**Contributors** RF is the review guarantor. The concept of the review was proposed by RF and JWG; the manuscript was drafted by NJO and RF, and edited by all authors. MH, NJO and RF designed the search strategy with advice from MM. NJO, MM and RF contributed to the development of the study eligibility criteria and data extraction criteria. JWG and CD provided expertise on type 1 diabetes. MM provided expertise on systematic review methodology. NJO and RF screened the titles and abstracts of records retrieved from the searches using the predetermined inclusion criteria, with any doubt or disagreement discussed with JWG. NJO and RF independently completed data extraction and assessed the quality and risk of bias of the included studies using the Newcastle-Ottawa Scale (NOS), consulting JWG if any disagreements. DK provided expertise on data extraction and analysis. All authors read, edited and approved the final manuscript.

**Funding** This research was supported in part by the MRC grant MR/N015428/1.

**Competing interests** RF has received a grant from the Medical Research Council MR/N015428/1 for his work as principal investigator of the project 'Investigating the inter-relationship between diabetes and children's educational achievement'.

**Patient consent for publication** Not required.

**Provenance and peer review** Not commissioned; externally peer reviewed.

**Data availability statement** All data relevant to the study are included in the article or uploaded as supplementary information.

**ORCID iDs**

Natalie Jayne Oakley http://orcid.org/0000-0003-3943-8723
Dylan Kneale http://orcid.org/0000-0002-7016-978X
Robert French http://orcid.org/0000-0002-9064-9721

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
