## [Reviewer comments · BMJ Open]

ARTICLE DETAILS

TITLE (PROVISIONAL)	Type 1 Diabetes Mellitus and Educational Attainment in Childhood: A Systematic Review
AUTHORS	Oakley, Natalie; Kneale, Dylan; Mann, Mala; Hilliar, Mariann; Dayan, Colin; Gregory, John; French, Robert

VERSION 1 – REVIEW

REVIEWER	Matthew Cooper Telethon Kids Institute, Australia
REVIEW RETURNED	14-Aug-2019

GENERAL COMMENTS	Overall the paper is well-written and gives, broadly, a nice overview of the literature in this area, rightfully addressing the paucity of good contemporaneous studies. The challenge I have is it is hard to be blown away by the paper, perhaps because the objective was so specific. “high stakes” “standardised testing” “end of compulsory schooling”. I think most readers who are even loosely across this area are likely to have picked, on reading the objective alone, that the two Swedish papers would be the only two to be included. That said, as duly noted, it is important to highlight the need to more (contemporaneous) research in this area, with globally diverse representation. I have minor comments to make in the aim of improving the paper. Minor comments - Unclear what ‘gradient’ refers to towards end of results section and in Table 3 – I can’t see how, in the context of the results summarised, this is any different to ‘mean difference’?- Page 12 line 7 – it is unclear exactly what comparisons the 0.11 to 0.20 relates to, please clarify- The second limitation could be repackaged. I don’t think it was excluding early papers as a feature of the design is much of a limitation (it’s 2019 after all), I think this point should be more focused on how the authors are forced to draw conclusions from papers where children were treated under (potentially) old management regimens and tested under different testing conditions. The discussion bounces between these two distinct points, I think the latter is the more important one here.
---

REVIEWER	Bushra Khokhar University of Calgary
REVIEW RETURNED	19-Sep-2019

GENERAL COMMENTS	Please avoid switching between first person and third person and also please be consistent and chose weather you owant to write in present or past tense. Introduction  Line 13, do not abbreviate IDF without spelling it out completely the first time. Please add a ouple sentences on what high stakes testing is. Line 58 - please list out the four home nations. Methods:  Page 6, line 14, what do mean by continuous outcome? Limitations: Line 39, do not abbreviate NICE without spelling it out the first time Conclusion:  Line 25, The studies included in this review.
--

REVIEWER	Linda Beckman Karlstad University
REVIEW RETURNED	10-Oct-2019

GENERAL COMMENTS	The aim with this paper was to evaluate studies on whether diabetes has an impact on educational attainment and school attendance. The topic is important. This is very well written paper which clearly follows the official guidelines for systematic reviews. The tables and figures are clear. The main concern with the results are that the data in the included studies was collected long ago and that the medical health care had advanced substantially after that, which probably make it easier for children with diabetes to keep up with school work. However, the authors address this concern in a transparent way. Beside this, I really don't have anything to add.
--

VERSION 1 – AUTHOR RESPONSE

Reviewer: 1

Reviewer Name: Matthew Cooper

Institution and Country: Telethon Kids Institute, Australia

Please state any competing interests or state 'None declared': None declared

Please leave your comments for the authors below

Overall the paper is well-written and gives, broadly, a nice overview of the literature in this area, rightfully addressing the paucity of good contemporaneous studies.

The challenge I have is it is hard to be blown away by the paper, perhaps because the objective was so specific. "high stakes" "standardised testing" "end of compulsory schooling". I think most readers who are even loosely across this area are likely to have picked, on reading the objective alone, that the two Swedish papers would be the only two to be included.

That said, as duly noted, it is important to highlight the need to more (contemporaneous) research in this area, with globally diverse representation.

I have minor comments to make in the aim of improving the paper.

We are very grateful for your comments and agree that a principal limitation of our review was the use of narrow inclusion thresholds resulting in the small number of included studies. We have ensured to discuss this fully in the limitations section of the review, alongside thoughts for future research, as we agree that it is vital to obtain further more contemporaneous research in this important area.

Minor comments

- Unclear what 'gradient' refers to towards end of results section and in Table 3 – I can't see how, in the context of the results summarised, this is any different to 'mean difference'?

Text within the results section on page 8, Table 3 on page 8, Summary of Findings table on page 9 and conclusion on page 12 has been amended with 'gradient' removed for clarity.

- Page 12 line 7 – it is unclear exactly what comparisons the 0.11 to 0.20 relates to, please clarify. This relates to the difference found in mean grades when Dahlquist looked at different age groups of diagnosis. For clarity the text has been amended to include the original data, as below. Dahlquist estimated conditional means for age groups of diagnosis (<2, 2-5, 5-10, 10-15 years), reporting the lowest mean grades in children diagnosed before two years of age (2.97 +/- 0.09 vs 3.08-3.17 in children diagnosed at an older age), however this difference was found to be not significant.

- The second limitation could be repackaged. I don't think it was excluding early papers as a feature of the design is much of a limitation (it's 2019 after all), I think this point should be more focused on how the authors are forced to draw conclusions from papers where children were treated under (potentially) old management regimens and tested under different testing conditions. The discussion bounces between these two distinct points, I think the latter is the more important one here.

We found this a particularly interesting point and have made sure to amend this section in order to highlight this more effectively.

The second limiting restriction was excluding publications before 2004 in view of changes in the treatment of T1DM. The 2015 National Institute for Health and Care Excellence (NICE) guidelines(36) state that since 2004 there have been major changes in routine management of type 1 diabetes. Whilst this is pertinent to the UK and most Western countries, this date was perhaps less appropriate for developing countries. In addition, studies published since 2004 were based on cases from earlier years before those changes, for example, the cohorts for the included studies in this review were born 1972-85. This may impact the current applicability of conclusions drawn from these studies, as children with T1DM are likely to now have very different treatment regimens. Both included studies also highlight the change in the school marking system in Sweden in 1998 from a five-level numerical scale to a four-level alphabetical system. Changes to other general examination conditions over this time should also be considered, both in terms of national standardised testing and the use of administrative data records. This shows that more contemporaneous studies are required to assess the impact of advances in treatment alongside current examination conditions and data records. For example, Cooper et al.(7) includes data for children born up to 2003, and so can report insulin pump therapy being associated with better school performance than twice daily injections.

Reviewer: 2

Reviewer Name: Bushra Khokhar

Institution and Country: University of Calgary, Canada

Please state any competing interests or state 'None declared': No competing interests.

Please leave your comments for the authors below

Please avoid switching between first person and third person and also please be consistent and choose whether you want to write in present or past tense.

Thank you for highlighting this, the manuscript has now been checked and amended accordingly throughout.

Introduction

1. Line 13, do not abbreviate IDF without spelling it out completely the first time.

Sentence amended accordingly in the introduction on page 4.

2. Please add a couple sentences on what high stakes testing is.

A small paragraph incorporating this has been added in the introduction on page 4.

3. Line 58 - please list out the four home nations.

Sentence amended accordingly in the introduction on page 5.

Methods:

1. Page 6, line 14, what do mean by continuous outcome?

Thank you for highlighting this. We were making the distinction between continuous measures of attainment (such as a point score or percentage) with binary attainment measures (graduating high school vs not). However, we agree that this is not necessary and we have now removed this phrase and amended the paragraph accordingly.

Limitations:

Line 39, do not abbreviate NICE without spelling it out the first time

Sentence amended accordingly in the 'Limitation of review' section on page 11.

Conclusion:

1. Line 25, The studies included in this review.

Sentence amended accordingly in the conclusion on page 12.

Reviewer: 3

Reviewer Name: Linda Beckman

Institution and Country: Karlstad University, Sweden

Please state any competing interests or state 'None declared': None declared

Please leave your comments for the authors below

The aim with this paper was to evaluate studies on whether diabetes has an impact on educational attainment and school attendance. The topic is important. This is very well written paper which clearly follows the official guidelines for systematic reviews. The tables and figures are clear. The main concern with the results are that the data in the included studies was collected long ago and that the medical health care had advanced substantially after that, which probably make it easier for children with diabetes to keep up with school work. However, the authors address this concern in a transparent way. Beside this, I really don't have anything to add.

We are very grateful for your comments and agree this is a very important area of research, with more studies required using more contemporaneous data in order to consider the substantial advances in management of T1DM.